# Nanostructured AlGaAsSb Materials for Thermophotovoltaic Solar Cells Applications

**DOI:** 10.3390/nano12193486

**Published:** 2022-10-05

**Authors:** Djamel Bensenouci, Boualem Merabet, Osman M. Ozkendir, Md A. Maleque

**Affiliations:** 1Faculty of Sciences and Technology, Mustapha Stambouli University, Mascara 29000, Algeria; 2Institute of Sciences and Technology, Salhi Ahmed University Centre, Naama 45000, Algeria; 3Computational Laboratory for Hybrid/Organic Photovoltaics (CLHYO), Istituto CNR di Scienze e Tecnologie Chimiche “Giulio Natta” (CNR-SCITEC), Via Elce di Sotto 8, 06123 Perugia, Italy; 4Department of Natural and Mathematical Sciences, Faculty of Engineering, Tarsus University, Tarsus 33400, Turkey; 5Faculty of Engineering, International Islamic University of Malaysia (IIUM), Jalan Gombak 53100, Malaysia

**Keywords:** thermophotovoltaics, nanostructured materials, AlGaAsSb, XAFS spectroscopy, DFT, Botlztrap

## Abstract

Thermophotovoltaic conversion using heat to generate electricity in photovoltaic cells based on the detraction of thermal radiation suffers from many engineering challenges. The focus of this paper is to study the nanostructure of AlGaAsSb for thermophotovoltaic energy conversion using lattice-matched heterostructures of GaSb-based materials in order to overcome the current challenges. The XAFS spectroscopy technique was used to analyze electronic structures and optical properties of GaSb, (Al, In) GaSbAs. The XAFS spectroscopy analysis showed a powerful decay at peak intensity that reveals to be related to a loss in Sb amount and light As atoms replaced in Sb atoms by 25%. Moreover, it was found that Al/In doped samples have highly symmetric data features (same atomic species substitution). The narrow direct bandgap energy, E_g_ of Al_0.125_Ga_0.875_Sb_0.75_As_0.25_ material raised (0.4–0.6 eV) compared to conventional photovoltaic cell bandgap energy (which is generally less than 0.4 eV) with weak absorption coefficients. The thermoelectric properties of AlGaAsSb computed via Botlztrap code showed that the electrons made up the majority of the charge carriers in AlGaAsSb. This nanostructure material exhibited a higher and acceptable figure of merit and demonstrated a promising thermoelectric material for solar thermophotovoltaic applications.

## 1. Introduction

The world of today is witnessing a rising demand for clean energies whose deployment may contribute to viable innovative energy initiatives, cut energy bills, and reduce warm greenhouses [1]. However, thermophotovoltaics (TPVs) that can be a solution to exploit solar energy (SE) in greening the world’s deserts, may know great breakthroughs, but power conversion efficiencies (PCE) have to be carefully rated [2].

As a highly efficient solid-state alternative to thermoelectric converters, TPVs directly convert electromagnetic radiation energy from thermal sources to electrical energy by photovoltaic (PV) effect, making difference with solar PVs asa source of radiation illuminating PV cells and controllable thermal emitters instead of the Sun. TPVs consist of high-temperature radiators, cavities including filters to control the infrared spectrum, and PV cells to convert thermal radiation into electricity, with advantages of conversion applicable to any high-temperature heat source, and enhanced efficiency by controlling the absorbed spectrum in PV cells compared to solar PV [2,3].

TPV cells consist of layers fabricated from specialized materials that absorb sunlight, store it as thermal energy, and can then even emit this heat as light [3]. By configuring the material layers in different ways, the resulting light can be adjusted to specific wavelengths [4]. These TPV solar devices have potentially improved the performances of SE collection by converting broadband sunlight into narrow-band thermal radiation tuned for cells. Solar TPV gives a unique, solid-state approach to converting heat into electricity based on thermal radiation [5].

Over the past few decades, single-junction PV cell efficiencies have been developed, but are still far below the theoretical upper limit, driven by energy losses due to below band gap (E_g_) photons and hot-carrier thermalization, owing to a large solar spectrum distribution [6]. PV device concepts have been reported to reduce losses and enhance efficiencies, however all suffer from high manufacturing costs, complex device fabrication processes, and material instability and degradation [7,8]. Solar TPVs (STPVs), as a complementary alternative to conventional PVs based on irradiating cells with thermal radiation from heated emitters [9], are highly scalable for a wide range of power capacities, have no moving parts, and allow SE storage to generate electricity [10]. TPV cells, exploiting both light and heat, contain layers of specialized materials that absorb sunlight, store it as thermal energy, and even can emit this heat as light [11].

Moreover, STPV emitters are an excellent alternative in terms of efficiency, waste heat management, and portability [12]. Due to the relatively low temperature of emitters in STPV systems, low E_g_ PV cells are privileged to achieve a radiative heat transfer (HT) with greater efficiency [13]. However, the limitation of weak thermal radiation at frequencies larger than 1.1 eV indirect E_g_ of Si-based PV cells should be circumvented [14]. To overcome this shortcoming, low direct-E_g_ (0.5–0.8 eV) III−V semiconductors are used to convert all photons with energies of E ≥ E_g_, close to that corresponding to the irradiant peak in the emitter spectrum [15], as an attractive alternative to Si [15,16]. But few photons can be converted by 1.1 eV Si cells with limited performance [16], i.e., only those with energy slightly above the PV cell’s E_g_ can be converted efficiently into electricity, and high-energy photons rapidly lose their excess energy by thermalization of the generated electron–hole pairs [17].

On another hand, the efficiency of conventional PVs is limited by the thermionic emission of dark current densities and the distribution of hot electron energies, based on the fact that electricity is generated by converters from thermal energy in a power cycle based on the vacuum emission of electrons [18]. However, when used, heterostructures (HSs) have the effect to reduce such current by suppressing the generation and recombination of carriers in the depletion region, and by raising barriers to impede the diffusion current flow [19]. TPV cells, such as GaInAsSb/AlGaAsSb HSs of particular interest, offer a potential approach towards increasing the open-circuit voltage (*V*_OC_) by the presence of the wider-E_g_ side (of p-n hetero-junctions) of which materials owning can be used at heterointerfaces of heterojunction cells [20]. HSs for single-junction TPV cells, with a~0.53 eV E_g_ InGaAsSb base layer and wide-E_g_ AlGaAsSb window/cladding layers (closely lattice and wide-E_g_ GaSb substrates), have to be improved further in terms of performance [21]. For example, in [22] GaInAsSb/AlGaAsSb/GaSb STPVs with a 94% internal quantum efficiency (QE), a 70% fill factor (F_F_), and a 0.33 V V_OC_ with an AlGaAsSb window approach the theoretical limits, however, further increase in V_OC_ is expected to improve the overall TPV cell performance of such HSs [23].

As reported in the literature, performance enhancement of GaSb-based III-V applied in TPVs, temperature dependence of their E_g_ and total efficiency [5,24], and TPV energy conversion improvement in such systems [1,12,25], had been completed. Although GaSb substrates suffer from high costs, small sizes, non-availability of semi-insulating wafers, and poor thermal conductivity of deleterious effect on some high-power devices [26], AlGaAsSb/GaInAsSb epilayers [27], quantum well structures [28] and microstructures, Ref. [29] had been nevertheless grown. GaSb-based TPV HSs have exhibited high external QEs in the mid-IR range, which helps recover waste heat from high-temperature blackbody sources [30]. In terms of thermal radiation transfer (TRT), however, the sub-wavelength gap and large temperature difference between emitters and PV cells while minimizing HT are not contributing to the photocurrent (I_PH_) generation. In this regard, numerous studies on TPV applications have been conducted.

Theoretically, Yu et al. [31] have shown that 2D rectangular gratings on GaSb surface serving as antireflection patterns for nano-E_g_ TPVs, should significantly enhance near-field (NF) radiative flux from emitters to GaSb cells, and improve output power and PCE. Experimentally, Inoue et al. [32] have shown ten-fold enhancement of I_PH_ in PV cells, via an NF-TPV device with intermediate Si substrates that enable TRT suppression due to sub-E_g_ radiation by free carriers and surface modes, without degrading V_OC_ and Fermi level (F_F_). However, in thick emitters and doped substrate PV cells, a large amount of undesired HT in a frequency range below sub-E_g_ HT of PV cells has to be induced leading to a low PCE [21], since most of the thermal photons expected to excite electron−hole pairs reaching PV cells own energies less than 1.1 eV [14]. To boost PCE through carrier thermalization energy loss, hot carrier PV cells have to be realized by capturing excess energy of excitinghot carriers for power generation by reducing the rate of carrier cooling [21]. Additionally, adding highly reflective rear mirrors on PV cells maximizes the extraction of luminescence boosting voltages, and reflecting low-energy photons back into emitters, recovering the energy and improving the TPV efficiency [33].

Besides, a variety of problems experienced by energy users can be solved by using renewable microgrids. The provision of power in remote areas is expensive, due to difficult access, low population density, and harsh climate. Hence, it is decisive to deploy systems based on SE in such isolated areas to overcome shortcomings of electricity expensive or difficult and costly to maintain, or not friendly environment.

In the context of environmental concern, mitigating power loss in solar TPV systems due to dust is crucial when deploying solar systems in remote areas, which in turn may suffer from aerosol concentrations and sandstorms leading to the accumulation of layers of dust on solar arrays [34]. This could damage mainly GaSb HSs-based TPV solar module systems during rainfall, posing serious threats to such systems while converting energy. Strategies to replace Al, Au, and Ag metals in solar selective absorber coatings, to resist moisture, sand, dust, and similar phenomena, by nitrides of transition metals (TMs) have seen little success to date, while encapsulation approaches tend to compromise low-cost advantages of TPV absorbers. Coating with TMs such as nitrides coating to form a TMs-adsorbing layer (vulnerable to either saturation or contamination by rain or dust) on glass surfaces may help to reduce the risk of weakening the durability of solar absorber (SA) coatings, which is vital for TPVs. Using nitrides of IVB-to-VIB TMs (Cr, Ti, V, Zr, Nb, H_f_, Ta, Mo, and W) for SA coatings reveals that are a beneficial technique should be efficient to increase the optical coatings’ durability and expand the applicability of such alloys in optics [35,36].

For better STPV device implementation, waste heat and recycling low-energy photons of GaSb-based nanostructured materials could be potential candidates. Therefore, the present work presents theoretical results on TPV cells with extreme PCE at high emitter temperature, expecting a record for TPV efficiency, getting cells with top average reflectivity for below E_g_ photons, to recycle sub-E_g_ photons, and preventing solar coatings damages from sandstorms, dust, and rainfall in remote areas.

As well known, efficiencies of single-junction PV cells are still far below the theoretical limit of Shockley and Queisser, and a maximum of ~33.7% PCE at E_g_ ~1.34 eV (that can be overcome through tailoring metallic nanocomponents and chemical composition of absorbers), is dictated by energy losses due to below-E_g_ photons and hot-carrier thermalization [36,37]. To enhance TPV efficiency, spectral selectivity and high-temperature stability of absorbers and emitters should be improved [6], though thermal stability remains a challenge, since nanostructures become unstable at temperatures much below the melting point of the used materials [37]. At high operating temperatures, however, the reduced selectivity limits the effectiveness of nanophotonic emitters [38], which are privileged to selectively emit radiation above PV cell E_g_ [39].

Although PV cells owning selective absorption facilitate the recycling of low-energy photons and improve efficiency, the long-term thermal stability of nanostructured emitters has yet to be treated [40]. GaSb-based HSs that own selective emissions spectrum are suitable for TPV applications, thermally stable at high temperatures, and appear ideal when combined with III–Vs in (0.4–0.55) eV Eg-range, such as a~0.47 eV E_gInGaAsSb_ base layer on a~0.41 eV E_gAlGaAsSb_ window layer. The narrow E_g_ of such structures is prone to detrimental recombination effects and parasitic heating [41]. Therefore, PV cells HSs with typical E_g_ energy of 0.5–0.8 eV are required, since, for efficient electricity generation, the E_g_ of TPV cell should be controlled in the range of 0.3–0.7 eV [42]. Significant challenges involved in realizing near-field-truly rainbow trapping in TPVs are so that sub-wavelength E_g_ and large temperature differences between emitters and PV cells are required while minimizing HT. Hence, Materials for absorbers and emitters with long-term thermal stability in PV modules for high temperatures are so that HSs, composed of refractory metal nanometric films sandwiched between layers of dielectric materials deposited on bulk substrates, should be proposed. As well known, when much higher system efficiencies are theoretically achieved, parasitic losses become relatively lower.

However, the information on the thermoelectric (TE) and optical properties of AlGaAsSb nanostructured materials with reflectivity (by recycling low-energy photons to boost TPV efficiency) in literature is fearful [43]. Therefore, the aim was to study electronic structures and optical properties of AlGaSbAs nanostructured materials, that are compared (for some properties) to those of GaSb and InGaSbAs.

## 2. Computational Method

DFT band structure calculations [44] for nanostructured GaSb-based quaternary carried out within Wien2k (implementing the FP-LAPW method and GGA-PBE approximation) [45,46]. Full relativistic effects are calculated with the Dirac equations for core states, and the scalar relativistic approximation is used for the other states [47]. Calculations are performed using the highly accurate full potential projector augmented wave method [48], of potentials used to describe electron-ion interactions with three and five valence electrons respectively for ^13^Al(3s^2^3p^1^), ^49^In(5s^2^5p^1^), ^31^Ga(4s^2^4p^1^), ^33^As(4s^2^4p^3^), and ^51^Sb(5s^2^5p^3^), representing ionic cores. (In/Al)Ga/As/Sb atoms occupy the 8c/4a/4b/24e Wyckoff positions. To model, the 12.5% substitutional doping at different atomic sites, two (Al/In) atoms located at (¾, ¾, ¾) and (¼, ¼, ¼) positions are substituted by Ga atoms. Asatoms at (0, 0, 0) and (½, ½, ½) positions are substituted by Sb atoms.

Brillouin zone (BZ) integrations are performed with the tetrahedron method in a 2 × 2 × 2 Monkhorst–Pack *k*-pointmesh centered at high symmetry point Γ, inside BZ [49]. We adopted a cubic (*F-43m*#216) structure with 16 atoms in the primitive unit cell. Muffin-tin(MT) radii of In/Al, As, Ga and Sb were respectively equal to 2.4/2.52, 2.43, 2.48, and 2.45 Bohr. The values of R_MT_ × k_MAX_ parameter and separation energy are set respectively to 7 and −8 Ry. For accurate E_g_ with an improved band gap, Becke and Johnson’s potential and its modified versions havebeen employed. The present study reveals Al_0.125_Ga_0.875_Sb_0.75_As_0.25_nanostructured material with an E_g_ extremely suitable for STPV applications. For computing the optical properties, 4 × 4 × 4 shifted k-mesh grids were used for pristine and doped systems.

By optimizing lattice constants, band structures and densities of states are evaluated, and TE properties are obtained via the BoltzTrap code [50]. It interpolates the band structure obtained from the DFT calculations, do the necessary integrations (Fermi integrals) at different temperature and E_F_, and provides required transport coefficients as a function of temperature and carrier concentration. TE properties are computed using Boltzmann transport theory within constant relaxation time approximation and rigid band approximation. In terms of phonons dispersion, the lattice thermal conductivity is calculated using Slack’s equation (Equation (1)), given by [51]:

(1)$$\kappa_{l}=\frac{A\theta_{D}^{3}V^{1/3}m}{\gamma^{2}n^{2/3}T}$$
where *A* is a constant that can be calculated, *θ_D_* is the Debye temperature, *γ* is the Gruneisan parameter, *V* is the atomic volume, *n* is the number of atoms in the primitive unit cell, and m is the average mass of all atoms in the crystal.

For a different view ofthe electronic properties of the GaSb, Al_0.125_Ga_0.875_Sb_0.75_As_0.25_ and In_0.125_Ga_0.875_Sb_0.75_As_0.25_ semiconductors, XAFS calculations using the real space multiple scattering approach FEFF 8.20 code were performed [52]. This program reads instruction cards that are written in the input file for the computational steps providing electronic energy details, crystal data, and ambient conditions. That input files were created using the TkAtoms package, which is part of the IFEFFIT Shell interface [53]. In the input file, a gallium atom at the origin real space was chosen as the source atom for both GaSb, Al_0.125_Ga_0.875_Sb_0.75_As_0.25_, and In_0.125_Ga_0.875_Sb_0.75_As_0.25_ calculations. For the calculations, the input file of GaSb material in cubic (*F-43m*) geometry was created for the 10 Å thick cluster containing 159 atoms (Ga and Sb) with lattice parameters; a: 6.0940 Å. The second input file was prepared for a 10 Å thick Al_0.125_Ga_0.875_Sb_0.75_As_0.25_ material for Al substituted in 12.5% of Ga atoms and also As substituted in 25% of Sb atomic coordination. For the third step, In atoms were substituted in 12.5% Ga coordinations for the similar formula as In_0.125_Ga_0.875_Sb_0.75_As_0.25_.

## 3. XAFS Characterization

The electronic and crystal structure properties of Al_0.125_Ga_0.875_Sb_0.75_As_0.25_, In_0.125_Ga_0.875_Sb_0.75_As_0.25_, and GaSb semiconductors were theoretically studied by X-ray absorption fine structure (XAFS) spectroscopy technique. Calculations were performed with the code FEFF 8.20, which is a real-space multiple scattering approach (Figure 1a,b) [54]. XAFS spectra contain data on the electronic structure and crystals structure of the studied materials, and for this reason, data can be separated into two main regions for quality and in-depth work as X-ray absorption near-edge spectroscopy (XANES) and Extended-XAFS (EXAFS) [55]. XANES spectroscopy is sensitive to the inner shell ionization process observed in the XAS mechanism and contains rich information about the electronic structure of atoms in the materials studied, while the EXAFS region, consists of the tail portion where spectral fluctuations occur, indicative of interruption of travel between atoms with high kinetic energies and continuous scattering [56].

In Figure 2, Ga K-edge XANES spectra of GaSb, and (Al, In)_0.125_Ga_0.875_Sb_0.75_As_0.25_ are given in comparison. Ga K-edge spectra begin to rise at 10.366 keV and give a weak pre-edge structure at 10.375 keV.

Data within this region contains information about configurations of neighboring atoms, atomic distances, and coordination numbers.

K-edge absorption spectra of Ga atoms are a result of the transition of the excited 1s core electrons to the unoccupied 4p levels that are permitted according to the quantum selection rules. InGaSb, Ga atoms possess a Ga^3+^ oxidation state and 4s and 4p levels are totally empty. The only route for the excited 1s electrons is to the empty 4p levels, where 4s levels are dipole forbidden. Neighboring Sb atoms’ outer shells 5p are unoccupied, and in GaSb material Sb atoms become Sb^3−^. The electronic interplay seems to couple the 4s level of Ga and 5p level of Sb and built-up Ga 4s-5p (Sb) hybridized levels to where a small number of 1s electrons are located as a final state and give the pre-edge peak “A”. The main edge peak of Ga atoms was determined at 10.380 keV, resulting from the 1s–4p transition. All gallium K-edge peaks have a high agreement at the peak structure and values that confirm stable Ga electronic structure in materials. Beyond the main edges of Al_0.125_Ga_0.875_Sb_0.75_As_0.25_ and In_0.125_Ga_0.875_Sb_0.75_As_0.25_alloys, a tiny shift has been observed as a broadening in the band structure, which is related to the built metal bonding in Al and In substituted materials.

To study the crystal structure responses to Al, In and As substitution in GaSb nanostructure material, EXAFS scattering data are given in Figure 3, as a comparison. The scattering mechanism is a result of photoelectrons traveling among neighboring atoms where Ga atoms are the source atom, i.e., photoelectron emitter.

From k^2^-weighted curves of GaSb, Al_0.125_Ga_0.875_Sb_0.75_As_0.25_, and In_0.125_Ga_0.875_Sb_0.75_As_0.25_ alloys (Figure 3), all samples scattering intensities have a high agreement on peak structures, while Al and In doped Ga samples have slight shifts at high k-values. The shift is a result of the atomic response of GaSb crystals to Al, In, and Sb doping in the Ga environment. The shift addresses atomic displacements in Ga coordination due to the influence of different atomic species rather than Ga. The wavenumber varies oppositely to the wavelength, which means higher k-values closer to the source Ga atoms. The shift to the high k-values points out a disturbance in the crystal that has a press on the closest atomic shell around Ga atoms, i.e., Sb. Thus, the presence of heavy In atoms pushed Sb atoms away from Ga distances, while light Al atoms pull Sb atoms to closer distances, and fluctuating data occurred. However, the symmetry in scattering data confirms stable GaSb crystal structure even Al, and As doping. The probe of atomic positions around Ga atoms can be studied via the radial distribution of atoms that can be yielded with the Fourier transform of scattering data.

The FT-EXAFS scattering intensity data comparison, given in Figure 4, shows a high agreement for all samples for the peak positions. Although these peak positions have symmetry, peak magnitudes have slight changes. The peaks are related to the atomic positions of concerned atoms, which are assigned with atomic symbols, as given for GaSb. As for Sb peaks of GaSb, a powerful decay at peak intensity is related to the loss in the Sb amount and also light As atoms replaced in Sb atoms with 25%. As one type of atomic species substitution, both Al and In doped samples have highly symmetric data features. However, with the decrease in Ga atoms by 12.5%, Al or In due to the substitution, decays were also observed at about the same amount in the intensities. The magnitudes of Ga signals in the data have a narrowing for Al substitution, while a broadening in In substitution is observed, which is related to the atomic volumes of atoms that took place in the substitution process. The closest atomic position to the source Ga atom, which sits at the origin as the source atom and the photoelectron emitter, was 4 Sb atoms at a distance of 2.6388 Å as the first neighboring atoms shell, and the subordinations shell has 12 Ga atoms at a distance of 4.3091 Å.

## 4. Thermoelectric and Optical Properties

Photovoltaic and TE effects are linked via a complex relationship so the geometry of TPV devices is needed to be optimized. Besides, I_PH_ is mostly caused by the PV effect as opposed to the TE effects. Waste heat, as a very important renewable energy source to overcome the insufficiency of natural energies and global warming [57], is converted into electricity via the Seebeck effect [58].

Besides, the figure of merit (ZT as a function of the Seebeck coefficient, S) measures the performance of TE materials, so that a higher S (linked to a low amount of charge carrier) is needed for high TE performance [59]. Enhancing ZT has been a big challenge, and heavy atoms-based nanostructure such as GaSb hasbeen used to reduce thermal conductivities [60]. One of the efficient methods to increase ZT is reducing the dimensionality of the material, which increases S due to the increased density of states near E_F_ [61].

Although GaSb, InGaAsSb, and AlGaAsSb materials can be of interest for TE applications, AlGaAsSb experimentally synthesized, has been reported to own the highest band gaps [62] and lattice thermal conductivity [63], and it is required for efficient thermoelectric materials and predicted for good transport properties. The thermal conductivity (κ) is a parameter of interest to describe the heat conduction in terms of electrons/ phonons (κ*_e_*/κ*_l_*) within the crystal lattice, so a compound that has a lower thermal conductivity can be used for insulation purposes whereas the material having higher thermal conductivity is mostly used as heat sinks [64].

Light-matter interaction is decisive for the optical behavior ofmaterials for STPVs so that when thelightof suitable frequency interacts with such materials intra-band and inter-band transitions occur. Still, only the last ones are responsible for excitations and recombinations. Figure 5 and Figure 6 depict S and ZT of Al_0.125_Ga_0.875_Sb_0.75_As_0.25_ nanostructured material over the 0−600 °K temperature range, obtained from Boltzmann transport equation for electrons under a constant scattering time, and calculated by Boltztrap codevia relaxation-time *τ* dependent electrical conductivity (*σ*/*τ*) and electronic thermal conductivity (*κ**_e_*/*τ*). As the temperature increases, the Seebeck coefficient (Figure 5) decreases because of bipolar conduction. *Z* was also calculated as a function of the temperature in the same range, as shown in Figure 6, using the Seebeck coefficient and electrical and thermal conductivities.

Due to its semiconducting nature, Al_0.125_Ga_0.875_Sb_0.75_As_0.25_ nanostructured material exhibits a 0.268 ZT at absolute temperature, however, with temperature increasing (beyond 400 °K) it increases, and may reach high values in the range 600−1000 K. The narrowed E_g_ with high ZT suggests exploiting such nanostructured material for advanced STPVs.

In turn, a narrow direct E_g_ is a key parameter for STPVs. For this purpose, optical parameters (optical E_g_, absorption coefficient α, and optical conductivity σ) were primarily calculated to investigate the Al incorporation effects on HE. E_g_ energies for Al_0.125_Ga_0.875_Sb_0.75_As_0.25_ and In_0.125_Ga_0.875_Sb_0.75_As_0.25_ are found to be of 0.554 and 0.446 eV respectively, both of direct bandgap nature. E_g_ of (In,Al)GaAsSb matching GaSb, and emission wavelengths covering 1.5–4.5 μm for GaSb-based devices are key parameters to fabricate mid-infrared lasers in such wavelength range [65]. Figure 7 and Figure 8 illustrate thermal (κ) and optical (σ) conductivities, calculated respectively in the 0−600 °K temperature range and low temperature. Al_0.125_Ga_0.875_Sb_0.75_As_0.25_owns a σ that increases with increasing the photon energy over the 0–2.5eV range (Figure 8). When the thermal conductivity (Figure 7) is affected, it can give a high mobility and diffusion length of minority carriers [66] in the AlGaAsSb layer used for STPVs.

The reflectivity of AlGaAsSb is also calculated (Figure 9). At a lower energy regime, the reflectivity is recorded lower for such nanostructured material, but it was found to increase the middle order of the energy, and then decrease. The reflectivity of the AlGaAsSb nanostructured material is depending on the optical excitations among the energy bands appearing in the vicinity of E_F_.

The weak absorption coefficients α (Figure 10), indicating fractions of energy lost by the light wave while penetrating Al_0.125_Ga_0.875_Sb_0.75_As_0.25_ and In_0.125_Ga_0.875_Sb_0.75_As_0.25_ active layers, are the major challenges of GaSb-based HSs. This behavior may be attributed to the average lattice constant matched to GaSb, which can be explained by the electron-hole overlap [66]. The weak estimated absorption (limiting I_PH_ and PCE) near E_g_ may necessitate using rather thick nanostructured material. Although such a weak absorption enables enhanced materials consumption, the realization of efficient devices GaSb-based materials having modest transport lengths is possible when thicknesses are optimized.

## 5. Discussions

Expecting to avoid affecting the electronic structure significantly while aiming to improve the TPV performance of GaSb-based alloys, through alloying with a suitable element substituting, an XAFS spectroscopy study has been performed to analyze electronic structures and optical properties of GaSb, (Al, In) GaSbAs. Moreover, the TE performance/efficiency of these materials are judged here from ZT (unit-less quantity), which are calculated accurately with lattice thermal conductivity. Figure 5 and Figure 6 present the corresponding plots of S and ZT, as functions of temperature within the region of 0−600 K. Here, ZT as a function of temperature, computed for AlGaAsSb and presented in Figure 6, is such as at 50 K is it estimated to be 0.3, whereas at the temperature of 400 K it is 0.26, and then rises to 0.27 at 600 K. These values can reach the unity near 1000 K and show that AlGaAsSb has high thermo-power convergence efficiency. Moderate S and high ZT should dedicate to whether these compounds are good for TE applications or not. σ*_e_*, κ_e_/κ_p_, S, and ZT, the thermo-power convergence efficiency of AlGaSbAs, chosen to show reliability for STPVs, need to be all investigated. S, the TE parameter, that is a thermopower defined as a temperature gradient of potential across a material, informs on the heat conversion capability of a compound to electricity. Positive values of S, inversely proportional to the carrier concentration, throughout the temperature region confirm the p-type behavior AlGaAsSb. A decrease in S values with temperature could be attributed to the increase in carrier concentration. For TE efficient materials κ should be less to obtain high ZT. ZT of AlGaAsSbis much lower at 400 °K and estimated to be ~0.27 at 600 °K, can reach 0.6 at 1000 °K and beyond. ZT results of AlGaAsSb are in agreement with those of nanostructured Bi_2_Se_3_ and Ag_2_Te. Since κ may be reduced dramatically through alloying without affecting the electronic structure significantly, ZT may be then much higher and AlGaAsSb should be of extreme interest for STPVs for low-medium range temperatures. κ for the studied compound against temperature in the 0−600 °K shows that ZT values slightly reduced with temperature increase can be related to the decrease in S values and enhancement in κ near 600 °K and beyond. Besides, a reduction of κ can improve TE performance ofAlGaAsSb. κ curve shows an increasing trend with a rise in temperature. It increases from 0.192 W/mK at 200 °K to 6.51 W/mK at 600 °K.

The optical response of AlGaAsSb that should be examined for STPV applications is illustrated here by its reflectivity R (in%) (aiming to recycle low-energy photons), optical conductivity (σ), and absorption coefficient (α). R displayed in Figure 9 in the energy range of 0–20 eV shows the highest R of 65% corresponds to AlGaAsSb at its corresponding narrow E_g_. Tuning E_g_ of nanostructured Al_0.125_Ga_0.875_Sb_0.75_As_0.25_ via quantum size effects enables customizing solar cells’ absorption profile to match the sun’s broad spectrums.

The absorption of light by (Al,In)GaSbAs layers is so that the main picks of α for Al_0.125_Ga_0.875_Sb_0.75_As_0.25_ and In_0.125_Ga_0.875_Sb_0.75_As_0.25_ compounds appear as two new bands at 162, 188 cm^−1^ and 133, 244 cm^−1^ respectively, indicating that these compounds are coordinated to Al/In metal(III) centers. Although with weak α, uniform absorption can be assumed, and careful optimization of device parameters is needed to address the compromise between light harvesting and charge transfer, with the aim to achieve efficient thickness-insensitive STPV GaSb-based devices. A thickness improvement of such devices is essential for making full use of light, especially in semiconductors with relatively weak α. To achieve strong light absorption in active layers based on such compounds, textured surfaces, photonic crystals, nanowires, and gratings, as light trapping structures, should be used based on the excitation of surface plasmons and/or magnetic polaritons [67]. Although poor α of such compounds limit significantly spectral designs to enhance output powers and efficiencies [68] with suitable gaps between emitters and absorbers, such mediocre absorptions could be overcome, and conversion efficiencies should be enhanced [69]. Besides, the problem with the energy recirculation in such TPV systems is that materials with high emissivity should exhibit high absorption coefficients. Hence, it is required to tune the optical properties of GaSb-based HSs for strong emissivity [43]. High back-surface reflectance (Figure 9) that enhances the internal recycling of luminescent photons should recycle low-energy photons, and then boost TPV efficiency. However, it is mandatory to reach thermal stability and lower ohmic losses associated with high current densities, by developing multi-junction TPV cells, to reduce intrinsic trade-offs. For EH with STPV devices mitigating thermal losses is required for higher efficiency.

## 6. Conclusions and Way Forward

On the basis of XAFS investigation on the electronic structure of GaSb-based nanostructure material, and via DFT study, TE and optical properties of AlGaAsSb are assessed. The narrow direct E_g_ of Al_0.125_Ga_0.875_Sb_0.75_As_0.25_ material raised with weak absorption coefficients. The TE properties of AlGaAsSb, such as κ, S, and ZT, computed via Botlztrap code, have shown that according to S, electrons make up the majority of the charge carriers in AlGaAsSb. This nanostructured material having acceptable S and ZT could be a promising candidate forideal TE material for STPVs application.

As perspectives for future research, we expect to optimize, design, and construct efficient and cost-effective GaSb-based HSs for TPV applications, and treat long-term thermal stability nanostructured emitters. Since efficiency improvement with a rise in absorbers and emitters’ operating temperatures is beneficial to reach maximum efficiency, and narrowband emitters and heat conservation reaching are useful, it is hoped to tune thermal emission from nanophotonic structures and enable applications of near-field TRT of GaSb-based HSs to various domains such as EH and waste heat recovery. Designing multi-layered films from nitrides of TMs for SA coatings, and synthesizing GaSb TPV-based systems can be scalable for industrial use of TPVs that may offer a great advantage to developing advanced multifunctional coatings for energy-efficient and intelligent window applications.

## Figures and Tables

**Figure 1 nanomaterials-12-03486-f001:**
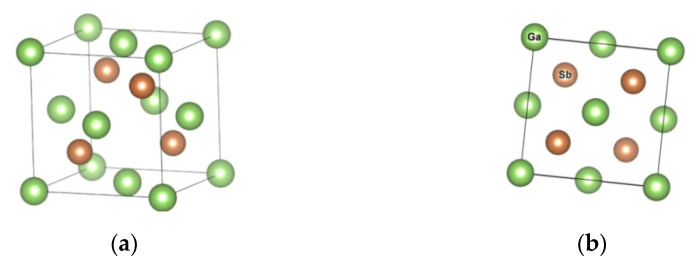
(**a**) 3D, (**b**) 2D view of GaSb compound.

**Figure 2 nanomaterials-12-03486-f002:**
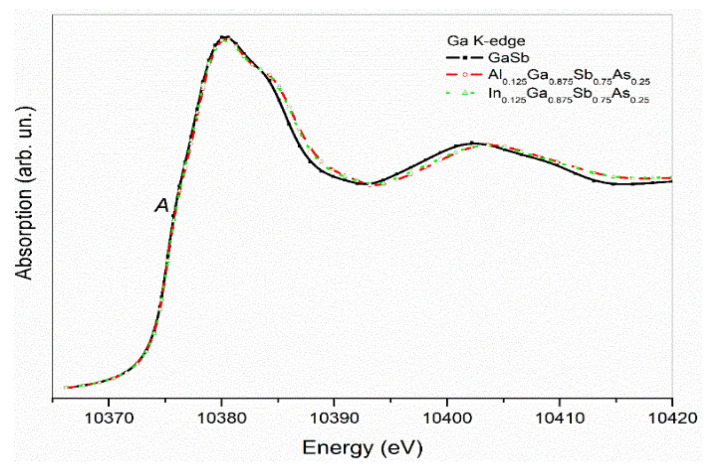
Ga K-edge XANES spectra of GaSb, (Al, In)_0.125_Ga_0.875_Sb_0.75_As_0.25_ nanostructured material.

**Figure 3 nanomaterials-12-03486-f003:**
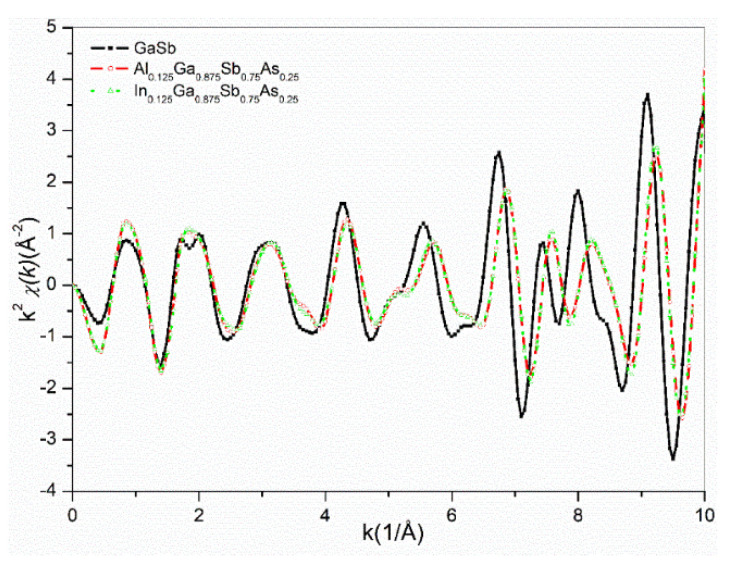
EXAFS scattering data of GaSb, (Al, In)_0.125_Ga_0.875_Sb_0.75_As_0.25_ alloys.

**Figure 4 nanomaterials-12-03486-f004:**
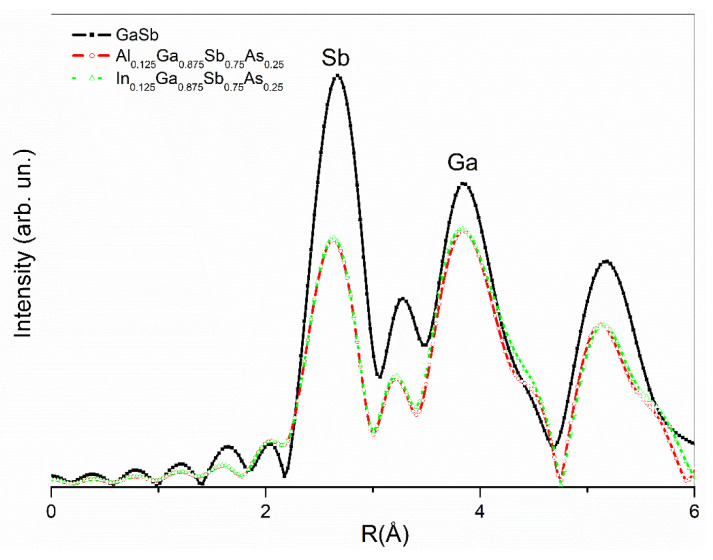
FT-EXAFS scattering data of GaSb and (Al, In)_0.125_Ga_0.875_SbAs nanostructured materials.

**Figure 5 nanomaterials-12-03486-f005:**
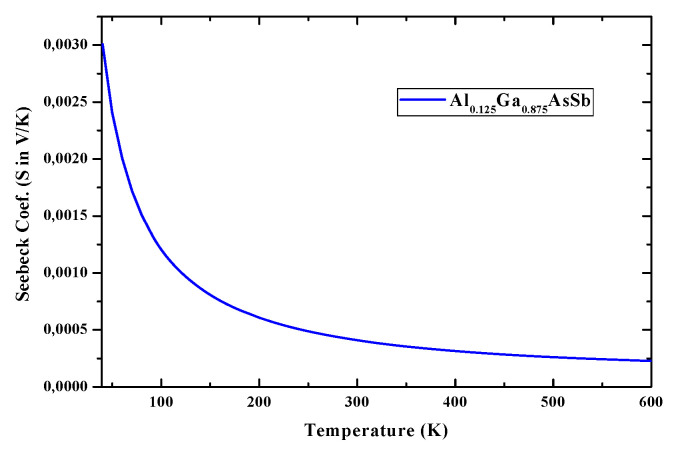
Seebeck coefficient (S) of AlGaAsSb nanostructured material within 0−600 °K range.

**Figure 6 nanomaterials-12-03486-f006:**
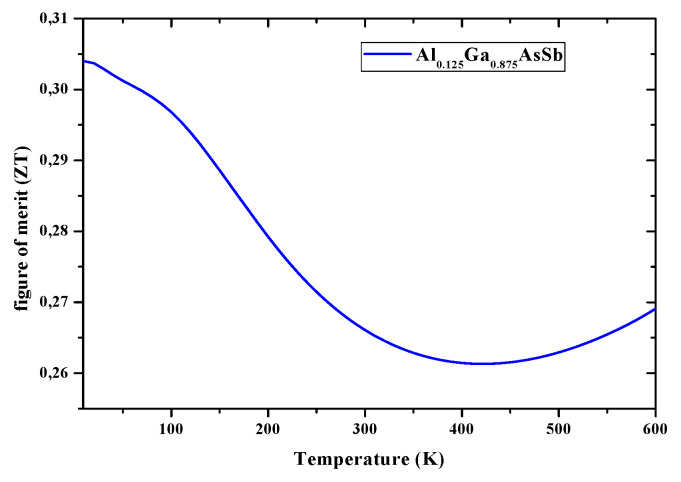
Figure of merit (ZT) of AlGaAsSb nanostructured material within the 0−600 °K range.

**Figure 7 nanomaterials-12-03486-f007:**
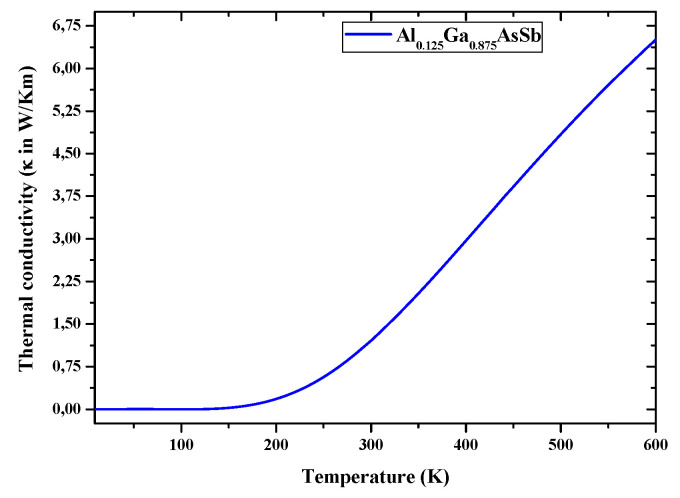
Thermal conductivity (κ) of the Al_0.125_Ga_0.875_Sb_0.75_As_0.25_ nanostructured material.

**Figure 8 nanomaterials-12-03486-f008:**
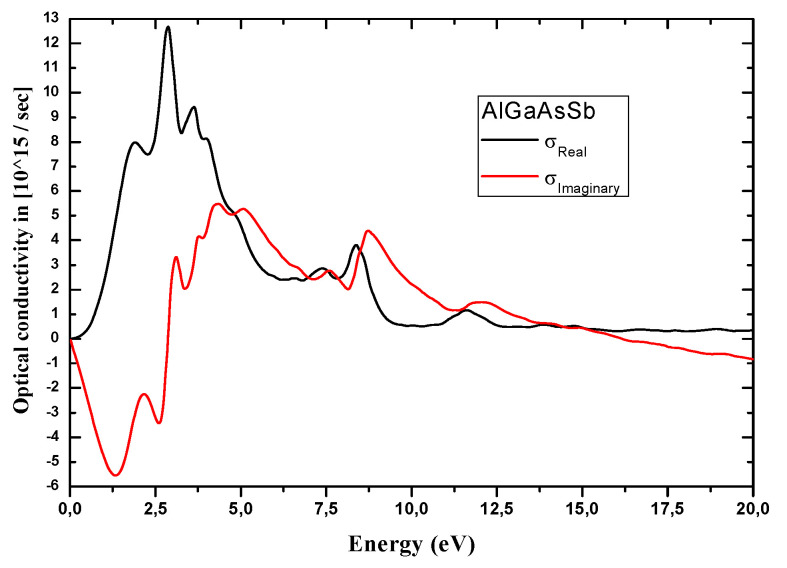
Real and imaginary optical conductivities of the nanostructured Al_0.125_Ga_0.875_Sb_0.75_As_0.25_.

**Figure 9 nanomaterials-12-03486-f009:**
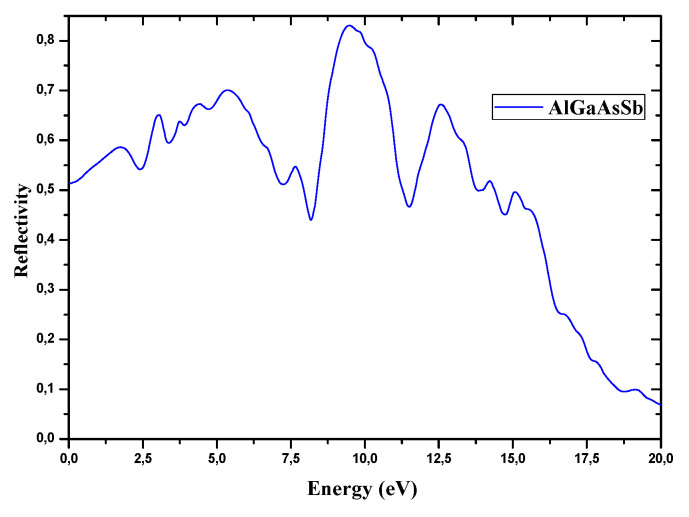
Reflectivity of the nanostructured Al_0.125_Ga_0.875_Sb_0.75_As_0.25_ within the 0−20 (and 35) eV energy range.

**Figure 10 nanomaterials-12-03486-f010:**
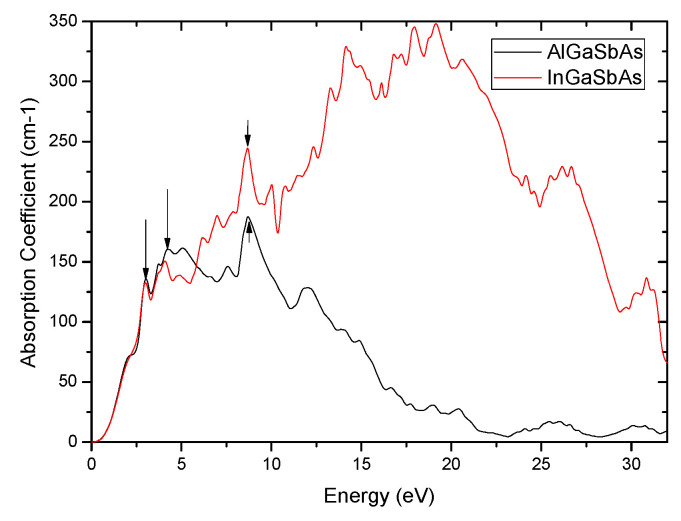
Absorption coefficient (essential parameter for TPVSs) of In_0.125_Ga_0.875_SbAs compared to that of the nanostructured Al_0.125_Ga_0.875_Sb_0.75_As_0.25_.

## Data Availability

The data presented in this study is available on request.

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
