# Peer review of "Nanostructured AlGaAsSb Materials for Thermophotovoltaic Solar Cells Applications"

_nanomaterials, 2022, doi:10.3390/nano12193486_

Round 1

Reviewer 1 Report

This study conducted by Bensenouci, et cl. focused on the thermophotovoltaic propery of the nano-structured AlGaAsSb. There are tons of results are obtained from the theoretical calculations and they correspond to the various figures in the manuscript. However, all the figures are organized in a very poor manner, like the figure quality is very low, all the size of texts in the figure are very random, and sometimes, even the important vertical axis is missing, Figure 4. All the figures must be properly revised before the acceptance of this paper. Besides, since the optical behavior is so critical for the photovoltaic application, the thermoelectric property should be important, especially it may have unfavorable role on the  photovoltaic conversion. The correlation should be precised classified for the current study. Overall, this paper is quite interesting and it can be accepted after moderate revisions. In the last, there a number of typos and grammar mistakes, which should be corrected.

Reviewer 2 Report

The manuscript “Nanostructured AlGaAsSb Materials for Thermophotovoltaic Solar Cells Applications” presents theoretical studies of AlGaAsSb nanostructure for thermophotovoltaic energy conversion using lattice matched heterostructures of GaSb-based materials. The goals and motivation of such a study are relatively straightforward. The manuscript is well-structured and presented.

The paper can be recommended to Nanomaterials.

Additional comments that need to be addressed:

1)     The sentence: ‘However, HSs, when used, have the effect to reduce such current by suppressing the generation and recombination of carriers in depletion region and by raising barriers to impede the diffusion current flow [19].’ should be ‘However, when used, HSs reduce such current by suppressing the generation and recombination of carriers in the depletion region and by raising barriers to impede the diffusion current flow [19].

2)     The sentence: ‘In the context of environmental concern, mitigating power loss in solar TPV systems due to dust is crucial when deploy solar system in remote areas which in turn may suffer from aerosol concentrations and sand storms leading to accumulation of layers of dust on solar arrays [34].’ should be ‘In the context of environmental concern, mitigating power loss in solar TPV systems due to dust is crucial when deploying the solar system in remote areas, which in turn may suffer from aerosol concentrations and sand storms leading to the accumulation of layers of dust on solar arrays [34].’

3)     The sentence: ‘The only route for the excited 1s electrons are to the empty 4p levels, where 4s levels are dipole forbidden.’ should be ‘The only route for the excited 1s electrons is to the empty 4p levels, where 4s levels are dipole forbidden.’

4)     The sentence: ‘The electronic interplay seem to couple the 4s level of Ga and 5p level of Sb and built up Ga 4s-5p (Sb) hybridized levels to where small number of 1s electrons located as a final state and give the pre-edge peak “A”. ‘ should be ‘The electronic interplay seems to couple the 4s level of Ga and 5p level of Sb and built up Ga 4s-5p (Sb) hybridized levels to where a small number of 1s electrons are located as a final state and give the pre-edge peak “A”.

5)     The sentence: ‘The main edge peak of Ga atoms were determined at 10.380 keV, which is a result of 1s-4p transition.‘ should be ‘The main edge peak of Ga atoms was determined at 10.380 keV, resulting from the 1s-4p transition.

6)     The sentence: ‘Light matter interaction is decisive for optical behavior of materials for STPVs, so that when light of suitable frequency interact with such materials intra-band and inter-bandtransitions take place, but only the last ones are responsible for excitations and recombinatons.‘ should be ‘Light-matter interaction is decisive for the optical behaviour of materials for STPVs so that when the light of suitable frequency interacts with such materials, intra-band and inter-band transitions occur. Still, only the last ones are responsible for excitations and recombination.

7)     The discussion does not compare the presented data with the known data from the literature. This part should be expanded and deeply discussed.

8)     In the introduction, PCE values obtained for GaSb and similar inorganic solar cells should be given. It would be comparative for the audience to mention the PCE of cells based on, e.g. GaSb and organic compounds. There are many examples of solar cells based on inorganic and organic compounds in the literature (Sol. Energy Mater. Sol. Cells, 191, 2019, 444-450.; Comput. Mat. Sci., 165C, 101-113, 2019.; Materials, 13(10), 2292, 2020.; Dyes Pigments, 200, 110166, 2022.; ACS Appl. Energy Mater. 2021, 4, 9, 9304–9314.; J. Mol. Str., 1207, 127771, 2020.; Sol. Energy Mater. Sol. Cells, 39, 1995, 11-18.; J. Phys. Chem. C, 126, 21, 8986–8999, 2022.)

Reviewer 3 Report

The paper is aimed on the description of thermal photovoltaic sun-light energy convertion.

The abbreviations (acronymes) should be removed form abstract. Beginnig of the introduction is too bizarre and decorative linguistically – should be corected.

Second paragraph in Introduction – the break before (E_g) is needed. If authors mentioned about limits for efficiency of solar cells, the Shockley–Queisser efficiency limit should be mentioned with indicating ways to overcome this limit in modern solar cells, Materials 2022, 15, 2254. In the same reference the short review of up to date efficiency list of various cell is provided, which is also worth to be cited. Simultaneously in this place the comparison of efficiency of thermal PV cell should be clearly stated. Though the Shockley-Queisser limit is mentioned in next part of the Introduction the form of Refs [37-39,42] should be avoided and accommodated to the style ‘one entry – one reference’. The phrase about 94% efficiency of thermal solar cells seems to be confusing and must be clarified. The same with regard to general idea of thermal solar cells – it should more precisely explained in the Introduction.

The phrase that  the efficiency of conventional PVs is limited by the thermionic dark current as well as by the distribution of hot electron energies must be clarified in more detail (“thermionic dark current” is not a best term here).

Italic should avoided in the Introduction.

“calculated using Slack’s equation(Eq. 1)” the break before (Eq. 1) is needed.

The paragraph “Computational method” should be developed -- it is too modest. 

The phrase “The electronic and crystal structure properties of, Al0.125Ga0.875Sb0.75As0.25, In0.125Ga0.875Sb0.75As0.25 and GaSb semiconductors were theoretically studied by x-ray absorption fine structure (XAFS) spectroscopy technique” sounds strange – theoretical study by x-ray spectroscopy technique?

Eq. (2) should be claried (derivation together with used symbols shoud be added).

Figs 1-3 should be clarified (how they were obtained) and captions must be developed.

The linguistic proofreading is recommended.

Round 2

Reviewer 2 Report

The authors satisfactorily addressed all points raised by myself and by other reviewers. The English style is still far from perfect. Please check the English carefully throughout the manuscript.

Please check the style of the references. In many places, they are given differently, e.g.:

[18] Shengxiang Wu.; Hogan N.; Sheldon M. "Hot electron emission in plasmonic thermionic converters." ACS Energy Lett. 2019 4(10), 2508-2513. should be [18] Wu.S.; Hogan N.; Sheldon M. "Hot electron emission in plasmonic thermionic converters." ACS Energy Lett. 2019 4(10), 2508-2513.

[21] Yi Z.; Conibeer G.; Liu S.; Zhang J.; Guillemoles J.-F.. "Review of the mechanisms for the phonon bottleneck effect in III–V semiconductors and their application for efficient hot carrier solar cells." Progress in Photovoltaics: Research and Applications 2022 30 (6), 581-596. Aneta S.; Zych D.; Szafraniec-Gorol G.; Gnida P.; Vasylieva M.; Schab-Balcerzak E. "Investigations of new phenothiazine-based compounds for dye-sensitized solar cells with theoretical insight." Materials 2020, 13 (10), 2292. Zych, D.; Aneta S.; Agata F. "Is it worthwhile to deal with 1, 3-disubstituted pyrene derivatives?–Photophysical, optical and theoretical study of substitution position effect of pyrenes containing tetrazole groups." Computational Materials Science 2019, 165, 101-113. Tournet, J.; Parola S.; Vauthelin A.; Montesdeoca Cardenes D.; Soresi, Frédéric Martinez S.; Lu Q.; et al. "GaSb-based solar cells for multi-junction integration on Si substrates." Solar Energy Materials and Solar Cells 2019, 191, 444-450. Sylwia Z.; Zych D.; Szafraniec-Gorol G.; Kotowicz S.; Malarz K.; Musioł R.; Slodek A. "Does the change in the length of the alkyl chain bring us closer to the compounds with the expected photophysical and biological properties?–studies based on D-π-DA imidazole-phenothiazine system." Journal of Molecular Liquids 2022, 120076.

should be

[21] Zhang Y.; Conibeer G.; Liu S.; Zhang J.; Guillemoles J.-F.. "Review of the mechanisms for the phonon bottleneck effect in III–V semiconductors and their application for efficient hot carrier solar cells." Progress in Photovoltaics: Research and Applications 2022 30 (6), 581-596. Slodek. A.; Zych D.; Szafraniec-Gorol G.; Gnida P.; Vasylieva M.; Schab-Balcerzak E. "Investigations of new phenothiazine-based compounds for dye-sensitized solar cells with theoretical insight." Materials 2020, 13 (10), 2292. Zych, D.; Slodek A.; Frankowska A. "Is it worthwhile to deal with 1, 3-disubstituted pyrene derivatives?–Photophysical, optical and theoretical study of substitution position effect of pyrenes containing tetrazole groups." Computational Materials Science 2019, 165, 101-113. Tournet, J.; Parola S.; Vauthelin A.; Montesdeoca Cardenes D.; Soresi, Frédéric Martinez S.; Lu Q.; et al. "GaSb-based solar cells for multi-junction integration on Si substrates." Solar Energy Materials and Solar Cells 2019, 191, 444-450. Zimosz S.; Zych D.; Szafraniec-Gorol G.; Kotowicz S.; Malarz K.; Musioł R.; Slodek A. "Does the change in the length of the alkyl chain bring us closer to the compounds with the expected photophysical and biological properties?–studies based on D-π-DA imidazole-phenothiazine system." Journal of Molecular Liquids 2022, 120076.

Reviewer 3 Report

the resubmission is not corrected according to the previous recommendations

in particular, it is not allowed to include multiple references to one entry 

the manuscript must be corrected according to previous recommendation before any evaluation for publication 

Round 3

Reviewer 3 Report

I do not recommend the publication of the submission, unless the Authors thorougly fulfill all previous suggestions. 

The manuscript is still not carrefully written. 

A thorough proofreading is needed.

The references must be corrected.  They must  comply with the style: one entry one  paper. 

The submission must be again revised according previous recommendation. 
